# Thermal Properties of Geopolymer Based on Fayalite Waste from Copper Production and Metakaolin

**DOI:** 10.3390/ma15072666

**Published:** 2022-04-05

**Authors:** Aleksandar Nikolov, Alexandar Karamanov

**Affiliations:** 1Institute of Mineralogy and Crystallography, Bulgarian Academy of Sciences (IMC-BAS), Acad. G. Bonchev Str., bl.107, 1113 Sofia, Bulgaria; 2Institute of Physical Chemistry, Bulgarian Academy of Sciences, Acad. G. Bonchev Str., bl.11, 1113 Sofia, Bulgaria; karama@ipc.bas.bg

**Keywords:** geopolymer, fayalite, copper slag, thermal, fire-resistance

## Abstract

In the present study, thermal properties of geopolymer paste, based on fayalite waste from copper producing plants and metakaolin, were analyzed. The used activator solution was a mixture of sodium water glass, potassium hydroxide and water with the following molar ratio: SiO_2_/M_2_O = 1.08, H_2_O/M_2_O = 15.0 and K_2_O/Na_2_O = 1.75. High strength geopolymers pastes were evaluated after exposure to 400, 800 and 1150 °C. The physical properties (absolute and apparent density, water absorption) and compressive strength were determined on the initial and the heat treated samples. The phase composition, microstructure and spectroscopic characteristics were examined by XRD, SEM-EDS, FTIR and Mössbauer spectroscopy, respectively. The structure of the heat-treated geopolymers differs in the outer and inner layers of the specimens due to variation in the phase composition. The outer layer was characterized by a reddish color and more rigidity, while the inner core was black and less viscous at elevated temperatures. The results showed that geopolymer pastes based on fayalite are fire-resistant up to 1150 °C. Moreover, after heat treatment at this temperature, the compressive strength increased by 75% to 139 MPa, while water absorption reduced by about 9 times to 1.2%. These improvements are explained with the crystallization of the geopolymer gel to leucite and K,Na-sanidine, and substitutions of Al/Fe in the geopolymer gel and iron phases.

## 1. Introduction

Geopolymers are synthetic inorganic polymer materials produced at a low temperature in alkaline or acidic media [1]. Ordinarily, a geopolymer structure is comprised of chains or networks of connected SiO_4_ and AlO_4_ tetrahedral units by sharing oxygen atoms. The negative electrical charge of an Al^3+^ ion is compensated by positive alkali cations provided by the activator solution [2]. The geopolymers are usually produced by two main components—a fine aluminosilicate precursor and an activator solution. The most common activator is a mixture of sodium water glass and potassium or sodium hydroxide solution. On the other hand, the paramount geopolymer precursor is metakaolin—a calcined kaolinite clay, which was used for the first time by Joseph Davidovits in the breakthrough synthesis of geopolymers in the late 1970s [1]. Since then, the geopolymer technology has evolved to a wide range of potential raw materials. The research is focused on unutilized appropriated industrial, agriculture and municipal waste, such as fly and bottom ash, slag, tailings, red mud, biomass ash, sludge, construction waste, glass waste, agricultural waste, etc., [3,4,5,6]. The homogenization of the suited activator solution and aluminosilicate precursor yields geopolymer paste with properties similar to ordinary Portland cement (OPC) paste and, in particular cases, even superior. The geopolymer materials possess good fire resistance [7,8,9,10,11,12,13], chemical resistance [14,15,16,17,18,19,20], high strength [21,22], rapid hardening [23,24], and considerable adhesive strength to concrete, metal and glass [25,26]. Moreover, geopolymers are a sustainable alternative to OPC due to significantly lower CO_2_ emissions [27]. The Portland cement industry is responsible for about 8% of global CO_2_ emissions, of which approximately 55% are emitted from the decarbonization of limestone and 40% are the result of heating the kilns to about 1450 °C [28]. Contrary, geopolymer production is characterized by a lack of calcium carbonate decarbonization and a significantly lower activation temperature—about 750–850 °C for metakaolin production, compared to OPC.

One of the main drawback of OPC is the reduced fire-resistance due to the loss of mechanical strength at high temperatures, resulting in damage or total collapse. The residual strength of OPC-based concrete does not exceed 20–30% after firing at 800–1000 °C due to the dehydration and destruction of the C-S-H gel and other crystalline hydrates [29]. At high temperatures and/or high heating rates, a spalling effect is observed, which is a detrimental problem in Portland cement-based materials [11,30]. On the other hand, geopolymers have an intrinsic fire-resistance, possessing chemical stability, low deformation and spalling resistance on exposure to elevated temperatures [31]. There has been extensive research on the thermal properties of geopolymers [9,10,32,33,34,35,36,37]. The superior thermal properties result from the hardening mechanism of the geopolymer materials and the absence of hydrated phases. The water in the geopolymerization process ensures the workability and contact of the reactants, then it is released during the polycondensation stage [38]. The product of geopolymerization is a three-dimensional aluminosilicate network which is supported by alkalis [2]. Alkalis are also presented in traditional ceramics. During the production of ceramics, feldspars, or other alkali containing raw materials, act as the flux in the temperature range 1100–1300 °C. Practically, the formation of the liquid phase with alkalis governs the densification processes. This melt, vitrified at cooling, leads to an increase in the amount of amorphous phase in the products. At a low temperature, the traditional geopolymer is characterized mainly by an amorphous structure. However, with the increase in temperature, the geopolymer starts to recrystallize, mainly into plagioclases, feldspathoids or feldspars, which then, similarly to traditional ceramics, melt, forming a liquid phase. The main mineral phases after heating geopolymer systems are sodalite, nepheline, albite, kaliophilite, leucite, sanidine, anorthoclase, kalsilite, cristobalite, etc., [1,39]. The result of geopolymer exposure to elevated temperatures is a quasi-ceramic structure with excellent service properties and a significant level of structural stability, without or with limited explosive spalling [7]. A potassium-based geopolymer formulation exhibited significant fire-resistance up to 1300 °C [2,40]. The potential applications of fire-resistant geopolymer materials are: concrete with improved fire-resistance, mainly for tunnel construction; non-fired masonries for high temperature equipment; fire resistant panels or protective coatings as fire barrier; high-temperature adhesives and composites; etc.

The properties of a hardened geopolymer greatly depend on the starting raw materials. There is a vast amount of industrial waste potentially applicable as precursors. Part of these raw materials are hardly marketable or even unusable because of the presence of undesirable elements, such as iron, magnesium, etc., and/or hazardous heavy metals. A particular case is the usage of iron-rich waste. The importance of the iron content on the high-temperature performance of a geopolymer based on fly ashes was highlighted by Rickard et al. [34,41,42]. The presence of iron oxides directly affected the thermal properties by influencing the thermal expansion, altering the phase composition and changing the morphology after heating. Although the presence of iron plays an important role in the structure and properties of geopolymers, the role of iron is not fully studied. Moreover, Davidovits stated that iron can enter into polymer chains forming chains of ferro-sialate type -O-Si-O-Fe-O-Al- [43]. Geopolymers based on industrial iron-rich raw materials have been studied by: Komnitsas et al., examining Greek and Polish ferronickel slag [44,45,46], and Onisei et al. [47]—Belgian slag and synthetic slag from a pilot plasma reactor.

Our preliminary studies demonstrated the potential of fayalite waste (a Bulgarian iron-rich industrial by-product from copper production) to be alkali-activated [48,49]. Furthermore, high strength geopolymers were prepared with the addition of metakaolin. The thus obtained geopolymer paste showed a remarkable compressive strength of over 100 MPa [50]. The aim of the present study is to elucidate the thermal behavior and the related structural changes, phase transformations and physical properties of the high-strength geopolymers based on fayalite slag and metakaolin. To the authors’ knowledge, there is no data about fire-resistant properties of geopolymers based on iron-rich slags from heavy metal production.

## 2. Materials and Methods

### 2.1. Geopolymer Precursors

The main geopolymer precursor in the present study is fayalite waste, a copper production by-product from the biggest plant in southeast Europe—Aurubis AG, located in Pirdop, Bulgaria. Fayalite slag, also named as iron-silicate fines, is a fine powdery material—product of flotation process of slag from flash furnace and converters. The chemical composition is presented in Table 1. The mineral phases comprised in the slag are fayalite, magnetite and pyroxene [49]. The particle size of the fayalite waste is under 200 μm, with 50% under 20 μm. The fayalite waste was dried at 100 °C to constant mass prior to usage.

In order to increase the properties of final geopolymer, commercial metakaolin was used as secondary precursor. The metakaolin was produced by calcination of local pure kaolinite clay (provided by “Kaolin Inc.”, Senovo, Bulgaria). The wet residue of the metakaolin on 45 μm sieve was 0.40%. Chemical composition was characterized mostly by silicon and aluminum oxides with minor trace elements (Table 1).

A binary activator solution containing both potassium and sodium cations was prepared by dissolving solid pellets KOH (Sigma–Aldrich) in mixture of tap water and sodium silicate solution with molar ratio (SiO_2_/Na_2_O = 2.98) and density of 1.42 g/cm^3^. The activators were prepared one day before usage.

### 2.2. Methods of Analysis

The methods of analysis are summarized in Figure 1.

The chemical composition of the precursors was determined by XRF, performed at apparatus Rigaku Supermini 200 WD, Rigaku Corporation, Osaka, Japan, using pressed pellets.

The powder XRD patterns of the precursors and the geopolymer samples were performed with Philips PW1830 (Philips, Amsterdam, The Netherlands) and Cu Kα radiation with step size 0.05° and 1 s per step. 

The Mössbauer spectra were performed at room temperature by a WissEl electromechanical spectrometer (Wossemschaftlisiche Elektronik GmbH, Birkenfeld, Germany) working in a constant acceleration mode with 57 Co/Rh (activity at 10 mCi) source and α-Fe standard. The experimentally obtained spectra were fitted using CONFIT2000 software, version 4.12.26, Czech Republic [51]. The parameters of hyperfine interaction—isomer shift (δ), effective internal magnetic field (B), quadrupole splitting (ΔEq), line widths (Γexp), and relative weight (G) of the partial components in the spectra—were determined.

FTIR spectra were collected using a Tensor 37 spectrometer (Bruker, Billerica, MA, USA) with a 4 cm^−1^ resolution, 72 scans on KBr pellets in the spectral region of 400–4000 cm^−1^ at room temperature.

The cross-section images of specimens were performed with X-ray tomography apparatus Nikon XT H 225, Nikon Metrology, Leuven, Belgium.

The morphology and the chemical analysis of the samples were studied by scanning electron microscopy JEOL 6390 (Jeol USA, Inc, Peabody, MA, USA) with energy dispersive spectrometer (SEM-EDS). The fractures of samples were polished and covered with gold in vacuum.

The thermal behavior in the interval 20–1350 °C was evaluated at 5 °C/min by hot stage microscopy, HSM (Misura—HSM 1400, Modena, Italy, Xpert system solution), using cut geopolymer specimen with 6x6x6 mm^3^ dimensions.

Compressive strength was measured on cubic specimens with face area of 10 cm^2^ and polished top and bottom surfaces. For each series, three specimens were tested. The apparent density was calculated by measuring each dimension of the specimens with a digital caliper. The absolute density was estimated by gas pycnometer (AccyPy1330, Micromeritic, Norcross, GA, USA) after milling the samples below 26 μm. The relative porosity was calculated according to apparent and absolute density. The open porosity is equal to apparent density multiplied by water absorption. The water absorption was measured a day after heating procedures by soaking the samples in water until constant mass at laboratory temperature. 

### 2.3. Geopolymer Synthesis

The geopolymer recipe was based on our previous study [50]—series VFM4.5, characterized by standard Vicat consistence, according to EN 196-3:2016 [52]. Fayalite slag and metakaolin were mixed in weight ratio of 5:1 to obtain homogenous dry mixture. The activator was characterized by following molar ratios: SiO_2_/M_2_O = 1.08; H_2_O/M_2_O = 15.0 and K_2_O/Na_2_O = 1.75. The water to solid ratio of the prepared geopolymer was equal to 0.139 (*w*/*w*). The dried fayalite waste and metakaolin were homogenized with mechanical mixer for 30 s, activator solution was added and mixed for 60 s at low speed, followed by 30 s at high speed. The fresh mixture rested for 5 min was then mixed again for 30 s at high speed. The fresh geopolymer paste was poured into steel molds and covered with plastic to prepare cubic specimens with face area of 10 cm^2^. On the 2nd day, the specimens were demolded and kept in laboratory conditions until the 28th day (temperature—20 °C, relative humidity—60%).

## 3. Results

### 3.1. Hot Stage Experiments

Hot stage experiments were performed up to 1350 °C. The linear shrinkage and expansion of the sample is presented in Figure 2. A shrinkage of about 2% was observed at the interval 800–1150 °C, then the rate of densification increased and the shrinkage reached 4% at 1200 °C. At this temperature, signs of melting were also visible by the swelling at the base of the sample. It is interesting to note that the melting temperature of fayalite in similar waste is about 1160 °C [53]. Additional isothermal runs of two hour holdings were made at 850 °C and 1150 °C. No visible changes were noted at 850 °C, while at 1150 °C, a shrinkage of 4% was reached. The last results confirm the formation of the primary liquid phase due to the melting of fayalite, which provokes the densification process. With the subsequent increasing of the temperature and the amount of the formed liquid phase, an intensive expansion up to 30% at 1350 °C was observed. 

Rahier et al. [54] observed rather similar behavior of sodium-based metakaolin geopolymers, but in significantly lower temperatures. However, their sample started to shrink above the “glass transition” temperature, 650 °C, followed by expansion at about 750 °C. Rickard and Riessen [42] examined geopolymers based on fly ash and also observed a sharp shrinkage event followed by expansion in even lower temperatures—550–650 °C.

### 3.2. Thermal Treatment of the Prepared Geopolymer Specimens

The furnace experiments were performed in a muffle furnace. Specimens from the series VFM4.5 were heated up to 400, 800 and 1150 °C with a temperature ramp of 5 °C/min and a 1 h holding at the maximal temperature. Then the specimens were cooled down in a closed chamber. Each heating procedure was performed on a separate day, meanwhile the specimens were stored in the refrigerator to ensure an equivalent age.

#### 3.2.1. Physical and Mechanical Properties

Significant changes in the physical and mechanical properties of the geopolymers’ series VFM4.5 were observed after the furnace experiments up to 1150 °C. The color of the samples changed with the increase of the heating temperature from grey to red–brown due to the oxidation of the iron (Figure 3). A cross section of a specimen VFM4.5 exposed to 1150 °C reveal the depth of the oxidation (Figure 3a). A distinct boundary was observed between the inner and outer part of the specimen (Figure 3b). The outer layer (Figure 3d) showed a denser structure, which acted as a crust entrapping gases in the core. The increase of the temperature led to a coalescence of pores in volume due to a rising of internal pressure and the formation of the liquid phase. In fact, the presence of Fe^2+^ instead of Fe^3+^ led to a lower viscosity of the melt and lower melting temperatures. Thus, the inner core of the specimens was characterized by large rounded pores and caverns (Figure 3b,c). This is an evidence of having reached the temperature at which the viscous liquid phase in the inner core is formed. Pan and Sanjayan [55] examined an iron rich fly-ash-based geopolymer and observed viscous behavior on loading at temperatures above 680 °C.

The water absorption, densities, and the corresponding open and closed porosities of the prepared geopolymers after heating are presented in Table 2. The density of the samples gradually increased with the increasing of the heating temperature from 2.38 g/cm^3^ to 2.73 g/cm^3^. This densification of the material obviously correlates with the decrease of the relative porosity. Small cracks visible with bare eyes appeared in the specimens of the series VFM4.5-1150. At the same time, a certain decrease of the absolute density with the increase of the heating temperature was observed, with the largest drop between 400 and 800 °C. This behavior was probably related to the transformation of the fayalite phase and the resulting formation of the amorphous SiO_2_ phase with a significantly lower density. On other hand, in a similar experiment with geopolymers based on fly ash, the water absorption increased from 20% to 25% after exposure to 1150 °C [56].

The water absorption of the series VFM4.5 was about 10.7% and stayed at this value after heating to 400 °C. In further heating up to 1150 °C, the water absorption decreased significantly to reach only 1.21%, which corresponds to only 3% of open porosity. The densification of the samples with the temperature rise was related to a decrease in the relative porosity and the transformation of open porosity into closed. At 1150 °C, the open porosity was very low, whereas the closed increased to about 17 %. This behavior, as it is shown in Figure 2, was related with pore coalescence and is probably the result of the beginning of O_2_ release due to the reduction of Fe^3+^ in Fe^2+^. At a higher temperature (1200 °C), the reduction leads to the expansion of the samples (see Figure 2). The increase in the porosity with the temperature rise is reported by Rickard et al. [42] and Bakharev [10] on fly ash geopolymers. However, the expansion in their experiments takes place at a significantly lower temperature (800–900 °C), which highlights a reduced fire-resistance.

The initial geopolymers were characterized by an 80.1 MPa compressive strength (Figure 4). After heating to 400 °C, the strength increased by 48% to reach 118.2 MPa. Further heating to 800 °C led to a slight decrease of the compressive strength to 98.6 MPa. After heating to 1150 °C, the material increased its strength to 139.1 MPa, which is +74% compared to the initial compressive strength prior to heating.

The thermal behavior of geopolymers based on iron-rich fly ashes, classes C and F, was reported by Jiang [57]. Both series underwent a significant strength loss between 500 and 800 °C, and were destroyed after exposure to 1200 °C. Contrary, in the present study, a slight decrease in compressive strength was observed when the geopolymer was subjected to 800 °C, but further exposure to 1150 °C led to a higher compressive strength. The increase in compressive strength and low value of water absorption at 1150 °C was attributed to a sintering process during heating. Haddaji et al. [37] also reported a decrease in the strength of geopolymers based on metakaolin and phosphate sludge after treatment up to 600 °C, followed by a significant increase at temperatures above 800 °C due to the formation of new crystalline phases such as nepheline and carnegieite. 

#### 3.2.2. Powder XRD

Small fractions of each series were examined with powder XRD after heating (Figure 5). The results showed that heating the samples up to 400 °C did not bring phase changes. In the previous study [50], the TG analysis showed the start of the oxidation processes as being the increase in the mass of the samples at temperatures above 400 °C. Consistently, the samples heated at 800 °C showed the oxidation of Fe^2+^: phases of fayalite (2FeO·SiO_2_) and magnetite (FeO·Fe_2_O_3_) transformed into hematite (Fe_2_O_3_). This happened on the outer layer of the specimens where there was contact with oxygen from the atmosphere. To reveal this phenomenon, a sample of VFM4.5-1150 was separated into two samples—fractures (reddish in color) from the external outer layer, labeled VFM4.5-1150E, and from the inner core, labeled VFM4.5-1150I. Heating to 1150 °C in the presence of oxygen led to the transformation of fayalite and magnetite to hematite, while in the inner layers, fayalite and magnetite were retained. The melting of the pyroxenes and the dissolution of quartz was observed at 1150 °C. The subsequent crystallization of the geopolymer to a mixture of leucite and K,Na-sanidine was observed after heating at 1150 °C. Barbosa and MacKenzie [40] also observed partial recrystallization to leucite with a significant amorphous content retained in the potassium silate disiloxo geopolymer heated at 1200 °C.

#### 3.2.3. Mössbauer Spectroscopy

The Mössbauer spectrums and the parameters of the hyperfine interaction are presented in Figure 6 and Table 3. The Mössbauer spectrum of the raw fayalite slag (Raw-F) was composed of sextet and doublet components. The presented model, comprised of 3 sextets and 3 doublets, was used for spectrum fitting, as results for the calculated component parameters. The parameters of the sextet components correspond to the mineral magnetite: Sx1 is assigned to tetrahedrally coordinated Fe^3+^; Sx2 is assigned to octahedrally coordinated Fe^2.5+^—this is actually Fe^3+^ and Fe^2+^, but because of the fast electron exchange between them, the spectral effect is one sextet component; Sx3 is also assigned to octahedrally coordinated Fe^2.5+^ ions. The calculated parameters of the doublet components (Db1 and Db2) correspond to the two different positions of the Fe^2+^ ions in the structure of the mineral fayalite. The doublet component Db3 is related to the iron in the paramagnetic glassy state. Isomer shift values above 1.00 mm/s are typical for iron in the second oxidation state, thus it can be assumed that Db3 is related to Fe^2+^ ions in an amorphous phase.

The spectrum of the sample VFM4.5 was similar to the raw fayalite slag. There were not significant changes in the oxidation states or coordination of the iron after geopolymerization. This indicates that both magnetite and fayalite stayed predominantly inert or reacted only on its surface. However, part of the doublet component Db3 related to the amorphous phase was determined as 11% and decreased to 9% after geopolymerization, and so a new weak doublet named Db4 was observed. The calculated parameters (IS = 0.47 mm/s, QS = 0.76 mm/s) of Db4 correspond to the Fe^3+^ ions in the amorphous state, but the actual coordination of the iron is discussable. A similar doublet was reported by Pyes et al. [58] and Onisei et al. [47] after the geopolymerization of synthetic and industrial fayalite slag, respectively. They suggest that Fe^3+^ had an average 5-fold and 4-fold coordination with rather high Fe-O distances. Contradictorily, Mysen [59] stated that the upper limit for tetrahedral Fe^3+^ is IS = 0.25 mm/s, which is typical for Fe^3+^ in a tetrahedral coordination, whereas values above about 0.4 mm/s are related to Fe^3+^ in an octahedral coordination. Unfortunately, the observed doublet Db4 in the series VFM4.5 had a negligible relative weight—only 1%—and thus general conclusions could not be stated. The present results are identical to previous experiments where fayalite waste was alkali activated without the presence of metakaolin [48].

Substantial changes in the iron oxidation states in the outer layer of the specimens, where contact with air oxygen was possible, were observed after exposure to 1150 °C. The spectrum of VFM4.5-1150E significantly differs from other presented spectra. It is composed of two sextets and one doublet. The parameters of Sx1 are typical for Fe^3+^ ions in the structure of hematite, while Sx2 has a lower effective internal magnetic field (B) and wider line widths (Γexp), which could be due to the presence of defects in the crystalline structure of the hematite, or the possible partial inclusion of Al, Mg, etc. A doubled Db with an IS = 0.39 mm/s was registered, probably in the octahedral coordination. The broad line width (0.76 mm/s) implies that the iron inhabits in the amorphous state [60]. On the other hand, in the internal part of the specimen, an Fe^3+^ ion related to hematite was not detected. The doublets related to fayalite were reduced from a 46% to 37% relative weight of the partial component. The sextet component Sx3, related to the Fe^2.5+^ octahedral coordination (magnetite), increased significantly.

#### 3.2.4. FTIR

The FTIR spectra of raw fayalite slag, the geopolymer series VFM4.5 and the heat-treated samples are presented in Figure 7. Most of the peaks in the spectrum of the raw fayalite slag is related to mineral fayalite. The main bands located at about 870 cm^−1^ and 951 cm^−1^ are attributed to the asymmetric stretching vibrations of the ν3 mode in SiO_4_ [47]. The band at 825 cm^−1^ is due to the ν_1_ mode symmetric stretch [61]. The features in the region between 509 cm^−1^ and 476 cm^−1^ are related to the ν4 asymmetric bending vibrations of Si-O [43]. The band at 567 cm^−1^ is attributed to Fe-O stretching vibrations characteristic for magnetite. The band at about 635 cm^−1^ was pointed out by Pisciella et al. to be magnetite/pyroxene [62]. The shoulder band at 1040 cm^−1^ may be related to the Si-O-Si stretching vibration of pyroxene and/or an amorphous phase.

After geopolymerization, a broad new band appeared at 1016 cm^−1^ due to the stretching vibration of the T-O-Si bonds, where T is Si or Al. The band is characteristic for the geopolymers and its broadness is indicative of the long range disorder of the aluminosilicate network. New bands arose: at 694 cm^−1^—symmetric stretching vibration; and between 625 cm^−1^ and 462 cm^−1^—bending vibrations, all related to T-O-Si bonds [63]. The peaks correlated to fayalite and magnetite stayed evident also in the geopolymer spectra.

The heat treatment of the geopolymer sample brought different bands changes and shifts. The main band of the geopolymer sample at 1016 cm^−1^ blue shifted to a higher wavenumber at 1020 cm^−1^, revealing a lower degree of deformation of the silica network. Such behavior can be attributed to a structural rearrangement of the glass geopolymer network. The shift continued to 1027 cm^−1^ at 800 °C and 1150 °C. The bands related to fayalite in the geopolymer sample gradually decreased with the increase of the heat temperature. The band at 567 cm^−1^, attributed to the Fe-O stretching vibrations at temperatures above 800 °C, started to increase, correlating to the hematite formation (XRD results). The band shifted to 575 cm^−1^ at 1150 °C, differing from usual hematite band, which could be indicative of the structural incorporation of Al in the newly formed hematite.

#### 3.2.5. SEM-EDS

The structure of the geopolymer heated to 1150 °C (Figure 8) was characterized by heterogeneity, probably due to the reorganization and recrystallization of the phases observed in the XRD results. Images from the external layer (Figure 8a,c) and inner core (Figure 8b,d) are presented. Both zones contain spherical pores, but the pores in the inner core are significantly larger in size. The increasing of the pore size in the inner core is a consequence of coalescence due to the lower viscosity. At the external part of the heat-treated sample (Figure 8c), at point 3, we observed a newly formed hematite particle with a characteristic habit, which elemental composition revealed to be a partial Fe substitution of Al. This is supported also by the results of Mössbauer spectroscopy and FTIR. Point 2 and 4 are presented zones of geopolymer gel with only potassium and sodium as the alkali components, respectively. In Figure 8d, an image from the inner part of the heat-treated sample is presented. Herein, we observed Al substitution in the magnetite phases (point 2 and point 3). The gel phase (point 1) contained Ca, Na and K as balancing cations, and Fe, probably as a structural element. 

The SEM observation demonstrates comparable parts of crystal and amorphous phases. The presence of a significant amount of the amorphous phase explains the observed sintering process.

## 4. Conclusions

The conclusions of the presented study are summarized:
Geopolymers based on fayalite slag, a waste from a copper producing plant, showed fire-resistance up to 1150 °C. The thermal exposure of the obtained geopolymers led to significant changes in their physical properties and microstructure. Contrary to the ordinary Portland cement materials, the compressive strength of the obtained geopolymers increased from 80 MPa at room temperature to 139 MPa after heating to 1150 °C, and water absorption and open porosity decreased significantly to only 1.21% and 3%, respectively. Changes between the outer and inner layers of the specimens were observed above 800 °C. At 1150 °C, the structure of the outer layer was characterized by a color change to reddish due to the oxidation of the iron phases to hematite, while the inner core remained black due to the presence of magnetite. In both layers, there was partial crystallization of the geopolymer gel into leucite and K,Na-sanidine.At 1150 °C, the outer layer was more rigid, while the inner core was characterized by a lower viscosity due to the presence of Fe^2+^. As a result, coalescence processes in the inner core were observed.Partial substitutions of Al and Fe were detected in the geopolymer gel and magnetite/hematite phases after exposure to 1150 °C.At temperatures above 1200 °C, the geopolymers based on fayalite waste started to melt and expand vigorously. 


The fayalite slag showed a potential to be utilized for the production of geopolymer materials with superior properties—high strength (>80 MPa) and fire-resistance up to 1150 °C. Further studies are required to evaluate the chemical durability, size effect, etc.

## Figures and Tables

**Figure 1 materials-15-02666-f001:**
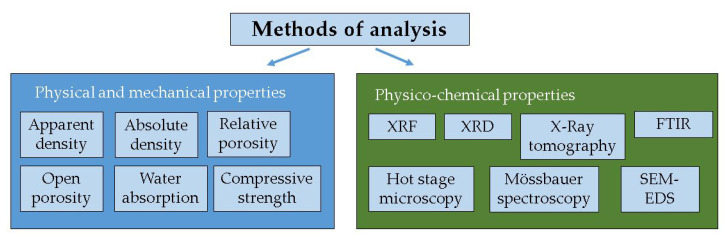
Methods of analysis of precursors and obtained geopolymers.

**Figure 2 materials-15-02666-f002:**
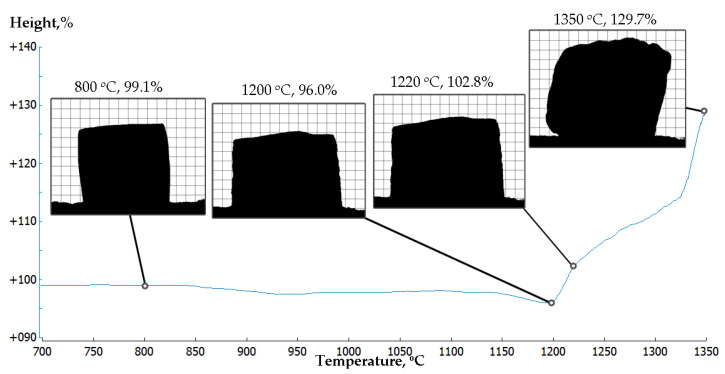
Hot stage experiment presents the changes in height of the specimens in the range of 700–1350 °C.

**Figure 3 materials-15-02666-f003:**
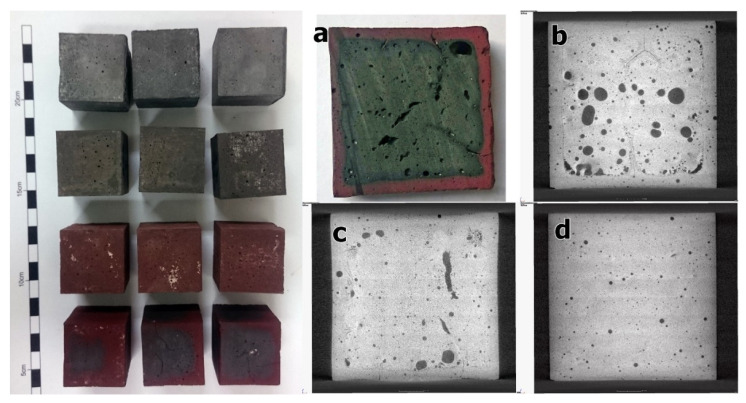
Geopolymer specimens (**left**)—from top to bottom—control series, 400 °C, 800 °C, 1150 °C. Cross section of geopolymer specimen series VFM4.5 heated to 1150 °C: (**a**) regular camera view; (**b**,**c**) X-ray tomography images in mid-part of the specimen; (**d**) X-ray tomography image close to surface in oxidation layer.

**Figure 4 materials-15-02666-f004:**
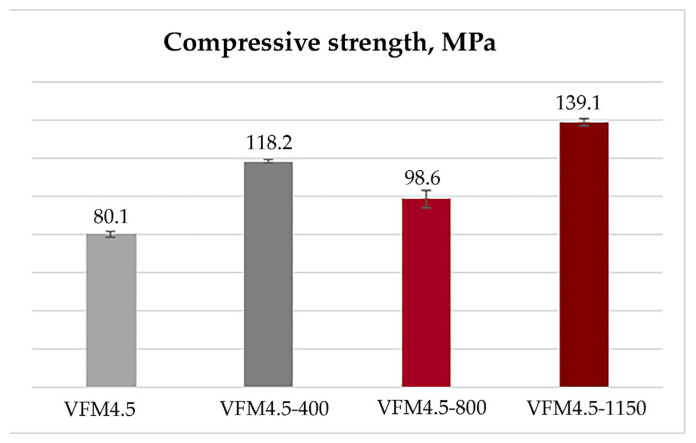
Compressive strength of the initial and thermally treated geopolymers series to 400, 800 and 1150 °C.

**Figure 5 materials-15-02666-f005:**
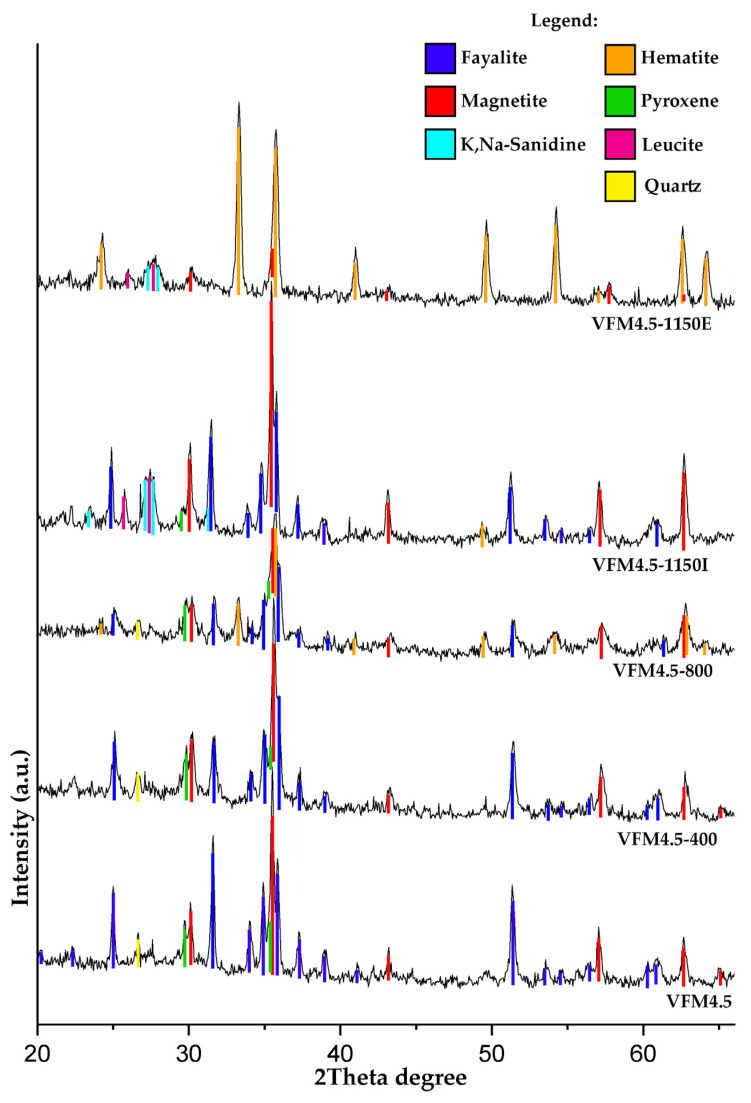
Powder XRD analysis of geopolymer series VFM4.5 exposed to temperatures from 400 to 1150 °C. Suffix: E—sample from external part of the specimen, I—sample from internal part of the specimen.

**Figure 6 materials-15-02666-f006:**
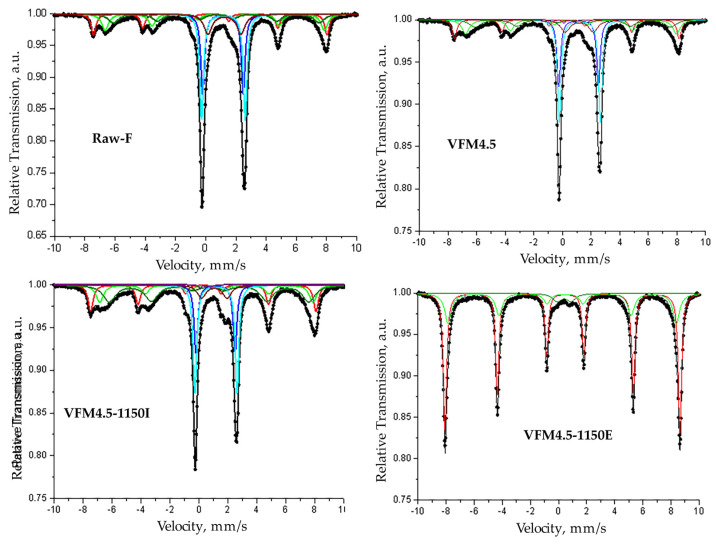
Mössbauer spectrums of geopolymer precursor (RAW-F), geopolymer series VFM4.5 and sample from interior and exterior part of VFM4.5 exposed to 1150 °C.

**Figure 7 materials-15-02666-f007:**
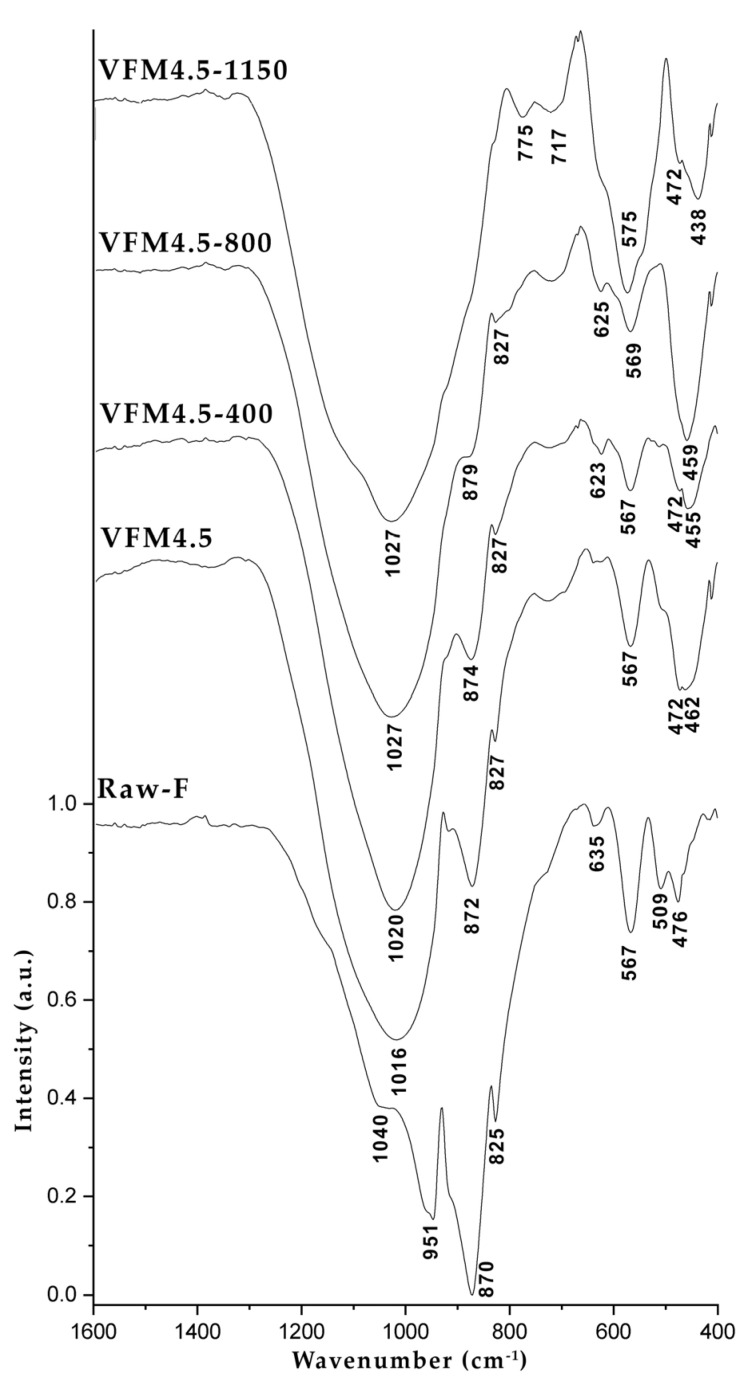
FTIR curves of raw fayalite, geopolymer series VFM4.5 and heat treated geopolymer at 400, 800 and 1150 °C.

**Figure 8 materials-15-02666-f008:**
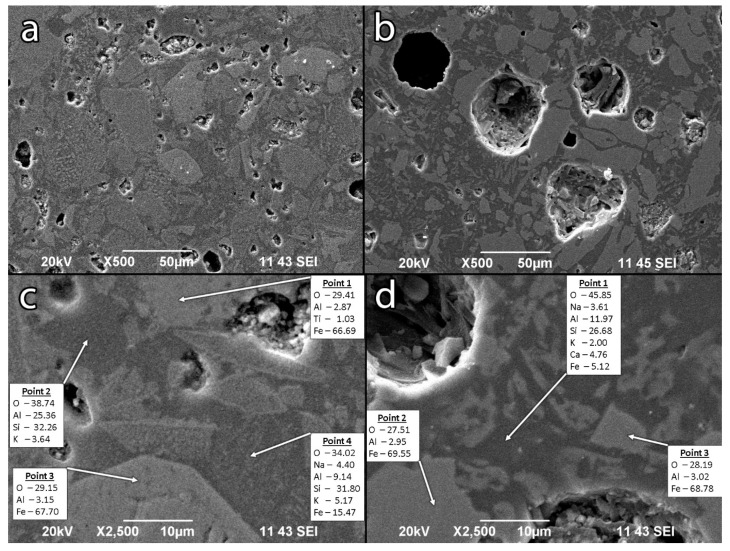
SEM-EDS images of geopolymer samples heat-treated to 1150 °C ((**a**,**c**) external part of VFM4.5-1150, (**b**,**d**) internal part of VFM4.5-1150). In boxes: elemental EDS analysis in %.

**Table 1 materials-15-02666-t001:** Chemical composition of the geopolymer precursors, determined by XRF, (%).

	Fe_2_O_3_	SiO_2_	Al_2_O_3_	CaO	ZnO	MgO	K_2_O	Na_2_O	CuO	PbO	TiO_2_	MoO_3_	SO_3_
Fayalite	58.42	29.34	4.40	2.66	1.32	0.89	0.71	0.58	0.49	0.37	0.30	0.27	0.26
Metakaolin	1.14	53.94	43.20	0.15	-	0.09	0.62	0.11	-	-	0.74	-	0.01

**Table 2 materials-15-02666-t002:** Physical properties of the initial and thermally treated geopolymers series to 400, 800 and 1150 °C.

Series	Apparent Density, g/cm^3^	Absolute Density, g/cm^3^	Relative Porosity, %	Water Absorption, %	Open Porosity, %
VFM4.5	2.38	3.59	34	10.7 ± 0.11	25
VFM4.5-400	2.49	3.58	30	10.6 ± 0.04	26
VFM4.5-800	2.61	3.44	24	7.12 ± 0.49	19
VFM4.5-1150	2.73	3.41	20	1.21 ± 0.44	3

**Table 3 materials-15-02666-t003:** Mössbauer parameters of geopolymer precursor (RAW-F), geopolymer series VFM4.5 and sample from interior and exterior part of VFM4.5 exposed to 1150 °C.

Sample	Components	IS, mm/s	QS, mm/s	B, T	Γ_exp_, mm/s	G, %
RAW-F	Sx1-Fe_3_O_4_, Fe^3+^ tetra	0.3	0	48	0.36	13
	Sx2-Fe_3_O_4_, Fe^2.5+^ octa	0.62	−0.05	45.1	0.54	15
	Sx3-Fe_3_O_4_, Fe^2.5+^ octa	0.72	−0.05	42.3	0.78	13
	Db1-Fe_2_SiO_4_, Fe^2+^—M1	1.14	2.68	-	0.3	21
	Db2-Fe_2_SiO_4_, Fe^2+^—M2	1.17	2.88	-	0.3	27
	Db3—Fe^2+^	1.24	2.19	-	0.65	11
VFM4.5	Sx1-Fe_3_O_4_, Fe^3+^tetra	0.3	0	48.8	0.37	13
	Sx2-Fe_3_O_4_, Fe^2.5+^octa	0.6	−0.02	45.8	0.62	17
	Sx3-Fe_3_O_4_, Fe^2.5+^octa	0.74	−0.05	42.5	0.91	14
	Db1-Fe_2_SiO_4_, Fe^2+^—M1	1.12	2.77	-	0.3	18
	Db2-Fe_2_SiO_4_, Fe^2+^—M2	1.21	2.87	-	0.3	28
	Db3—Fe^2+^	1.18	1.96	-	0.77	9
	Db4—Fe^3+^	0.47	0.76	-	0.3	1
VFM4.5-1150I	Sx1-Fe_3_O_4_, Fe^3+^ tetra	0.31	0	48.4	0.41	16
	Sx2-Fe_3_O_4_, Fe^2.5+^ octa	0.54	−0.04	45.8	0.61	14
	Sx3-Fe_3_O_4_, Fe^2.5+^ octa	0.7	−0.01	42.8	1.04	27
	Db1-Fe_2_SiO_4_, Fe^2+^—M1	1.15	2.71	-	0.3	14
	Db2-Fe_2_SiO_4_, Fe^2+^—M2	1.18	2.92	-	0.3	23
	Db3—Fe^2+^	1.07	1.76	-	0.71	6
VFM4.5-1150E	Sx1-α-Fe_2_O_3_, Fe^3+^ octa	0.38	−0.2	51.9	0.28	67
	Sx2-α-Fe_2_O_3_, Fe^3+^ octa	0.36	−0.2	50.4	0.56	28
	Db—Fe^3+^	0.39	1	-	0.76	5

## Data Availability

The data presented in this study are available on request from the corresponding author. The data are not publicly available.

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
