# Peer review of "Thermal Properties of Geopolymer Based on Fayalite Waste from Copper Production and Metakaolin"

_materials, 2022, doi:10.3390/ma15072666_

Round 1

Reviewer 1 Report

The manuscript is defiantly shown a great effort of experiments. However, I would like to address the following questions or comments to be taken into consideration when revising the manuscript:

1- Authors should improve the quality of all figures. The fonts in the figure should be consistent.

2- All chemical symbols should be in a standard format such as SiO4、Al2O4、CO2, please see in full text.. (for example SiO4、Al2O4、CO2?)

3- Materials and methods should be combined with graphics and text to increase the article readability.

4- Are the dimensions of the molds associated with standards?

5- Why the high-strength geopolymers pastes were exposed to 400, 800 and 1150 ⁰C? Authors must be given some standards.

6- All commercial names of devices are to be deleted.

7-The conclusion must be reinforced.

8- It is better to have this paper extensively edited to improve the language.

9- Authors must be correct references according to the journal guidelines.

10- The introduction part is not fully cited, and a lot of cement research has been carried out, for example, Jiajian Li, Shuai Cao, Erol Yilmaz. Compressive fatigue behavior and failure evolution of additive fiber-reinforced cemented tailings composites. International Journal of Minerals, Metallurgy, and Materials. 2022, 29(2): 345-355.

In summary, the reviewer believes that this manuscript may be accepted after the above strict revision through the abovementioned revisions.

Author Response

Thank you for your consideration of our manuscript entitled “Thermal Properties of Geopolymer based on Fayalite Waste from Copper Production and Metakaolin”. We have reviewed the comments of the reviewers and have thoroughly revised the manuscript. We found the comments helpful, and believe our revised manuscript represents a significant improvement over our initial submission.

 All new corrections are marked in yellow in the text and detailed responses to each reviewer are presented below. In response to the reviewers’ suggestions we have:

  • Improved the abstract.
  • Rewritten conclusions.
  • Reformatted the references according to MDPI guidelines and reinforced the references with 5 new sources.
  • Revised all figures and tables.
  • Improved the discussion part.
  • Corrected errors, mistakes and problems.

The manuscript is defiantly shown a great effort of experiments. However, I would like to address the following questions or comments to be taken into consideration when revising the manuscript:

Reviewer Comment:1- Authors should improve the quality of all figures. The fonts in the figure should be consistent.

Response: We agree with the reviewer’s suggestion. We have reformatted all figures.

Reviewer Comment:2- All chemical symbols should be in a standard format such as SiO4、Al2O4、CO2, please see in full text. (for example SiO4、Al2O4、CO2?)

Response: We have reformatted all chemical symbol to desired format.

Reviewer Comment:3- Materials and methods should be combined with graphics and text to increase the article readability.

Response: Thank you for this suggestion. We have added summarized graphic to increase the article readability.

Reviewer Comment:4- Are the dimensions of the molds associated with standards?

Response: The dimensions of the used molds are not according specific standard. The reason is that the standards for cement materials (EN 196-1) where the molds are prismatic 160x40x40 mm, is based on cement mortar – mixing the cement with fractioned sand. In our experiments we need simpler system thus we used only geopolymer pastes. Working with pastes we decided to use smaller molds to obtain cubic specimens with side area of 10 cm2. This information is already done in the text. Thank you for this question - we will take this as an advice for our future experiments.

Reviewer Comment:5- Why the high-strength geopolymers pastes were exposed to 400, 800 and 1150 ⁰C? Authors must be given some standards.

Response: To the best of our knowledge there is no such standards related to fire-resistance of geopolymers. However, we choose maximal temperature after previous published DTA/TG and the present results of the hot stage experiments where we observed significant expansion of the samples at higher temperatures.

Reviewer Comment:6- All commercial names of devices are to be deleted.

Response: We are not sure that we understand the question. According to our experience and journal practices the commercial information for the used apparatus usually is mandatory.

Reviewer Comment:7-The conclusion must be reinforced.

Response: We agree with the reviewer and found the proposition very valuable. We have rewritten the conclusion.

Reviewer Comment:8- It is better to have this paper extensively edited to improve the language.

Response: We agree with the reviewer and we did one more thorough language check on the entire manuscript. We also have taken into account the MDPI Manuscript English Editing advices.

Reviewer Comment:9- Authors must be correct references according to the journal guidelines.

Response: Sorry for our mistake, the references are revised according MDPI guidelines. 5 new references were added.

Reviewer Comment:10- The introduction part is not fully cited, and a lot of cement research has been carried out, for example, Jiajian Li, Shuai Cao, Erol Yilmaz. Compressive fatigue behavior and failure evolution of additive fiber-reinforced cemented tailings composites. International Journal of Minerals, Metallurgy, and Materials. 2022, 29(2): 345-355.

Response: We have added the particular article.

In summary, the reviewer believes that this manuscript may be accepted after the above strict revision through the abovementioned revisions.

Reviewer 2 Report

The paper "Thermal properties of geopolymer based on fayalite waste from copper production and metakaolin" is interesting and adheres to the scope of the journal Materials, after some corrections are necessary:
(1) The abstract needs to be better detailed, for example by adding complementary information about the mixtures and compositions analyzed, in addition to which properties of the geopolymers were effectively evaluated.
(2) The introduction should briefly address negative points and real and solid waste applications in geopolymer materials, add the following papers with this relationship: 10.1016/j.cscm.2021.e00662; 10.1016/j.cscm.2022.e00928; 10.1016/j.cscm.2021.e00792.
(3) The methodology section is very incomplete, it lacks information on the dosage of the mixtures and dosage method, in addition a table with information on the moistures and proportions should be added;
(4) There are several errors in posting references throughout the text, please check!
(5) Graphs should be redone, think about adding information on axes, improve discussions in terms of standard deviation and other information.
(6) The microstructural discussion can be improved by analyzing other studies in the current and international literature.
(7) Improve the conclusion, it should clearly show that you accomplished your proposed goals.

Author Response

Thank you for your consideration of our manuscript entitled “Thermal Properties of Geopolymer based on Fayalite Waste from Copper Production and Metakaolin”. We have reviewed the comments of the reviewers and have thoroughly revised the manuscript. We found the comments helpful, and believe our revised manuscript represents a significant improvement over our initial submission.

 All new corrections are marked in yellow in the text and detailed responses to each reviewer are presented below. In response to the reviewers’ suggestions we have:

  • Improved the abstract.
  • Rewritten the conclusions.
  • Reformatted the references according to MDPI guidelines and reinforced the references with 5 new sources.
  • Revised all figures and tables.
  • Improved the discussion part.
  • Corrected errors, mistakes and problems.

The paper "Thermal properties of geopolymer based on fayalite waste from copper production and metakaolin" is interesting and adheres to the scope of the journal Materials, after some corrections are necessary:
Reviewer Comment: (1) The abstract needs to be better detailed, for example by adding complementary information about the mixtures and compositions analyzed, in addition to which properties of the geopolymers were effectively evaluated.

Respond: We agree with the reviewer. We think that this information will be useful and we added additional information in the abstract.

Reviewer Comment: (2) The introduction should briefly address negative points and real and solid waste applications in geopolymer materials, add the following papers with this relationship: 10.1016/j.cscm.2021.e00662; 10.1016/j.cscm.2022.e00928; 10.1016/j.cscm.2021.e00792.

Respond:We agree with the reviewer’s suggestion. We found the studied waste geopolymer precursors interesting and we have added them.

Reviewer Comment: (3) The methodology section is very incomplete, it lacks information on the dosage of the mixtures and dosage method, in addition a table with information on the moistures and proportions should be added;

Respond: Thank you for your suggestion. We add more information about water to solid ratio. Now we think that this information is fair enough for a specialist to calculate and repeat the exact recipe.

Reviewer Comment: (4) There are several errors in posting references throughout the text, please check!

Respond: We are sorry for this mistake. There was a problem with the software. All figures and crossreferences are corrected.

Reviewer Comment: (5) Graphs should be redone, think about adding information on axes, improve discussions in terms of standard deviation and other information.

Respond: We reformat all figures with special attention for the font and sizes.

Reviewer Comment: (6) The microstructural discussion can be improved by analyzing other studies in the current and international literature.

Respond: We agree with the reviewer’s assessment of the microstructural discussion. We have reinforced the discussion parts.

Reviewer Comment: (7) Improve the conclusion, it should clearly show that you accomplished your proposed goals.

Respond: We agree with the reviewer and found the suggestion very valuable. We have rewritten the conclusion.

Reviewer 3 Report

The authors investigated the fayalite slag, waste from the copper-producing plants which could be utilized for the production of geopolymer materials with superior properties - high strength and fire-resistance resistance. The paper potentially contributes to the literature as it presents novel experimental results of interest for both research and practice purposes. However, the manuscript is affected by minor issues, and after minor revisions, it could be accepted for publication.

General report and comments:

  • Line 150. Please give a size of cubic specimens and reference to standard according to specimens were prepared.
  • Line 149, 150, etc. The superscript should be used for units description in the text.
  • Line 162. Please add a reference to the standard.
  • Line 167. Please add a reference to the standard for conditions of stored samples.
  • Section 3.1. Please align the text.
  • Section 3.2.1, 3.2.3. Please verify and correct the failed references.
  • Table 2. Please verify and correct the table according to Material Template requirements.
  • The first three figures have a number 1. Please verify and correct the numeration of figures and references in the text.
  • Conclusion chapter. Please, summarize the conclusions using bullet points. It would certainly emphasize the significance of the outcomes. Additionally, there should be closing remarks after the general conclusions (after the bullet points of conclusions), keeping in mind all the outcomes obtained.

Author Response

Thank you for your consideration of our manuscript entitled “Thermal Properties of Geopolymer based on Fayalite Waste from Copper Production and Metakaolin”. We have reviewed the comments of the reviewers and have thoroughly revised the manuscript. We found the comments helpful, and believe our revised manuscript represents a significant improvement over our initial submission.

 All new corrections are marked in yellow in the text and detailed responses to each reviewer are presented below. In response to the reviewers’ suggestions we have:

  • Improved the abstract.
  • Rewritten the conclusions.
  • Reformatted the references according to MDPI guidelines and reinforced the references with 5 new sources.
  • Revised all figures and tables.
  • Improved the discussion part.
  • Corrected errors, mistakes and problems.

Reviewer Comment: The authors investigated the fayalite slag, waste from the copper-producing plants which could be utilized for the production of geopolymer materials with superior properties - high strength and fire-resistance resistance. The paper potentially contributes to the literature as it presents novel experimental results of interest for both research and practice purposes. However, the manuscript is affected by minor issues, and after minor revisions, it could be accepted for publication.

General report and comments:

  • Reviewer Comment: Line 150. Please give a size of cubic specimens and reference to standard according to specimens were prepared.

Response: We agree with the reviewer that using standards is important. In our experiment we followed the mixture principles of EN 196-1. We didn’t reference the standard because we did not follow it strictly and didn’t use the same specimens specified in the standard. We have added more detailed information of the mixing procedure to 2.3. Geopolymer synthesis. The size of the specimens we specified by the face area of the prepared cubic specimens.

  • Reviewer Comment: Line 149, 150, etc. The superscript should be used for units description in the text.

Response: We agree and all units are checked and corrected.

  • Reviewer Comment: Line 162. Please add a reference to the standard.

Response: We have added reference to the standard.

  • Reviewer Comment: Line 167. Please add a reference to the standard for conditions of stored samples.

Response: We didn’t strictly follow particular standard. However, we have explained the procedure of storing samples.

  • Reviewer Comment: Section 3.1. Please align the text.

Response: We have checked and corrected the aligned in the text.

  • Reviewer Comment: Section 3.2.1, 3.2.3. Please verify and correct the failed references.

Response: We are sorry for this mistake. We have checked and corrected all references.

  • Reviewer Comment: Table 2. Please verify and correct the table according to Material Template requirements.

Response: We are sorry for this mistake. We have checked and corrected all tables.

  • Reviewer Comment: The first three figures have a number 1. Please verify and correct the numeration of figures and references in the text.

Response: The problem with the figures is corrected

  • Reviewer Comment: Conclusion chapter. Please, summarize the conclusions using bullet points. It would certainly emphasize the significance of the outcomes. Additionally, there should be closing remarks after the general conclusions (after the bullet points of conclusions), keeping in mind all the outcomes obtained.

Response: We agree with the reviewer and found the suggestion very valuable. We have rewritten the conclusion.

Round 2

Reviewer 1 Report

Now, the reviewer thought this revised manuscript can be accepted in its present form.

Reviewer 2 Report

The authors have made all indicated corrections, this paper can be accepted for publication.